# Polyamines Metabolism Interacts with γ-Aminobutyric Acid, Proline and Nitrogen Metabolisms to Affect Drought Tolerance of Creeping Bentgrass

**DOI:** 10.3390/ijms23052779

**Published:** 2022-03-03

**Authors:** Meng Tan, Muhammad Jawad Hassan, Yan Peng, Guangyan Feng, Linkai Huang, Lin Liu, Wei Liu, Liebao Han, Zhou Li

**Affiliations:** 1College of Grassland Science and Technology, Sichuan Agricultural University, Chengdu 611130, China; tanmeng194@163.com (M.T.); jawadhassan3146@gmail.com (M.J.H.); pengyanlee@163.com (Y.P.); feng0201@stu.sicau.edu.cn (G.F.); huanglinkai@sicau.edu.cn (L.H.); liulinsky@126.com (L.L.); lwgrass@126.com (W.L.); 2Institute of Turfgrass Science, Beijing Forestry University, Beijing 100089, China

**Keywords:** water balance, TCA cycle, photosynthesis, energy metabolism, nitrogen assimilation

## Abstract

Due to increased global warming and climate change, drought has become a serious threat to horticultural crop cultivation and management. The purpose of this study was to investigate the effect of spermine (Spm) pretreatment on metabolic alterations of polyamine (PAs), γ-aminobutyric acid (GABA), proline (Pro), and nitrogen associated with drought tolerance in creeping bentgrass (*Agrostis stolonifera*). The results showed that drought tolerance of creeping bentgrass could be significantly improved by the Spm pretreatment, as demonstrated by the maintenance of less chlorophyll loss and higher photosynthesis, gas exchange, water use efficiency, and cell membrane stability. The Spm pretreatment further increased drought-induced accumulation of endogenous PAs, putrescine, spermidine, and Spm, and also enhanced PAs metabolism through improving arginine decarboxylases, ornithine decarboxylase, S-adenosylmethionine decarboxylase, and polyamine oxidase activities during drought stress. In addition, the Spm application not only significantly improved endogenous GABA content, glutamate content, activities of glutamate decarboxylase and α-ketoglutarase, but also alleviated decline in nitrite nitrogen content, nitrate reductase, glutamine synthetase, glutamate synthetase, and GABA aminotransferase activities under drought stress. The Spm-pretreated creeping bentgrass exhibited significantly lower ammonia nitrogen content and nitrite reductase activity as well as higher glutamate dehydrogenase activity than non-pretreated plants in response to drought stress. These results indicated beneficial roles of the Spm on regulating GABA and nitrogen metabolism contributing towards better maintenance of Tricarboxylic acid (TCA) cycle in creeping bentgrass. Interestingly, the Spm-enhanced Pro metabolism rather than more Pro accumulation could be the key regulatory mechanism for drought tolerance in creeping bentgrass. Current findings provide a comprehensive understanding of PAs interaction with other metabolic pathways to regulate drought tolerance in grass species.

## 1. Introduction

Due to the increase in global greenhouse gases and ozone layer depletion over the last few years, extreme weather conditions such as continuous seasonal drought have brought severe challenges for the cultivation and management of crops worldwide. Drought stress reduces plant growth, water content, cell membrane stability, and also severely affects various physiological metabolisms, such as polyamine (PA) metabolism [1,2], γ-aminobutyric acid (GABA) metabolism [3], nitrogen (N) metabolism [4], and proline (Pro) metabolism in plants [5]. Metabolic disturbance and imbalance result in the inhibition of photosynthesis and respiration processes, leading towards reduced growth and development of plants. However, plants have developed a certain degree of self-resistance to adapt the changing environmental conditions at phenotypic, physiological, metabolic, and molecular levels [6].

PAs are low-molecular-weight aliphatic nitrogenous bases with strong biological activity. In plants, putrescine (Put), spermidine (Spd), and spermine (Spm) are three main PAs involved in response to unfavorable environmental conditions. In the past 50 years, PAs have generally been considered as a new class of plant growth regulator (PGR) because of their multiple biochemical and physiological functions related to plant growth, development, and adaptability to stress [7,8,9]. Plants can regulate the accumulation of Put, Spd, and Spm to enhance the tolerance to salt, drought, and cold stress [10,11]. It has been found that PAs content in cold-tolerant cucumber (*Cucumis sativus*) cultivar increased greatly, while cold-sensitive cultivar remained unchanged under low temperature stress [12]. For effects of PAs on drought tolerance in plants, foliar application of Spm could significantly improve the drought tolerance of rice (*Oryza sativa*) seedlings associated with alterations of leaf water status, photosynthesis, and membrane properties [13]. Exogenous Spm treatment significantly decreased drought-induced damage through enhancing antioxidant defense, carbohydrate metabolism, and dehydrins accumulation in white clover (*Trifolium repens*) [14,15]. Delayed chlorophyll (Chl) degradation regulated by the Spd was associated with changes in enzyme activities of Chl biosynthesis and catabolism in cucumber leaves under high temperature stress [16]. In addition, *Arabidopsis* mutant plants (*acl5*/*spms*) that could not synthesize Spm were more sensitive to drought stress. In addition, the Spm pretreatment could reverse the drought-sensitive phenotype of *Arabidopsis acl5*/*Spms* mutant that was unable to produce the Spm [17]. These results indicated that Spm regulated the tolerance against drought stress in plants by inducing a variety of changes in stress pathways.

Manipulation of PAs biosynthesis and metabolism could be an effective way to improve stress tolerance in plants. It has been found that salt stress improved enzyme activities of arginine decarboxylases (ADC) and ornithine decarboxylase (ODC) involved in PAs biosynthesis in salt-tolerant rice cultivar, but had no significant effect in salt-sensitive cultivar [18]. Exogenous GABA or chitosan (CTS) improved drought tolerance of white clover associated with enhancement of PAs synthesis and decline in PAs catabolism [15,19]. PAs also affected other metabolic pathways when plants responded to abiotic stress. For example, exogenous PAs treatment can enhance the accumulation of Pro in bermudagrass (*Cynodon dactylon*) under salt and drought conditions [20] and also induced the expression of *pyrroline-5-carboxylate synthetase 1* (*P5CS1*) which is a key gene involved in Pro synthesis in sour orange (*Citrus aurantium*) plants under oxidative and nitrosative conditions [21]. Exogenous Spd played an important role in activating nitrate reductase (NR) activity, thereby mitigating the inhibition of NR activity in wheat (*Triticum aestivum*) under drought stress [22]. Exogenous Spd decreased the toxic effects of ammonium nitrogen (NH_4_^+^) in cucumber seedlings via increases in the activities of glutamine synthetase (GS), glutamate synthase (GOGAT), and glutamate dehydrogenase (GDH) under Ca(NO_3_)_2_ stress [23]. In addition, diamine oxidase (DAO) performs a key role in the conversion of Put to GABA, and the application of aminoguanidine (AG), a DAO inhibitor, effectively inhibited the accumulation of GABA in the roots of German limonium (*Limonium tataricum*) [24]. Salt stress stimulated the DAO activity to improve accumulation of GABA in soybean (*Glycine max*) roots [25]. Under hypoxia condition, the DAO activity and PAs content increased significantly, which provided sufficient substrate for the formation of GABA in germinated faba bean (*Vicia faba*) [26]. However, it is still unclear how PAs metabolism interact with GABA, Pro and N metabolism to affect drought tolerance in plants.

Drought stress not only limits cultivation and utilization of horticultural crops, but also increases maintenance and management cost. Creeping bentgrass (*Agrostis stolonifera*) is one of the most important cool-season perennial species of the Poaceae family grown worldwide as a turf or lawn in public parks due to its fine texture and strong regeneration ability after pruning. Creeping bentgrass is also used widely in sports turf such as golf course, bowling green, and lawn tennis because it can form high-quality turf [27]. Previous studies have shown that creeping bentgrass is vulnerable to a variety of abiotic stresses during growth and development, including high temperature [28], saline alkali stress [29], and drought [30]. The objectives of this study were to elucidate whether the Spm improved drought tolerance of creeping bentgrass through regulating PAs metabolism and to further explore potential effects of PAs metabolism on Pro metabolism, N absorption and assimilation, and GABA metabolism under drought stress. Current findings will provide a comprehensive understanding of PAs interacting with other metabolic pathways to regulate drought tolerance in plants.

## 2. Results

### 2.1. Effects of Spm Pretreatment on Water Status, Membrane Stability, and Photosynthesis

To examine whether exogenous application of Spm affected photosynthesis and water status under drought stress, relative water content (RWC), electrolyte leakage (EL), and photosynthetic parameters were detected. Phenotypic changes showed that drought stress inhibited plant growth and accelerated leaf wilting (Figure 1A). After 12 days of drought stress, RWC decreased by 44% or 51% in stressed plants with or without the Spm application, respectively (Figure 1B). Exogenous Spm significantly alleviated the decrease in leaf RWC under drought stress (Figure 1B). Under drought stress, the EL increased significantly, but the EL of plants pretreated with Spm decreased by 17% at 12 d and 22% at 18 d when compared with plants without the Spm pretreatment, respectively (Figure 1C).

Under drought stress, total Chl, Chl a, Chl b, photochemical efficiency (Fv/Fm), performance index on absorption basis (PI_ABS_), net photosynthetic rate (Pn), transpiration rate(Tr), stomatal conductance(Gs), and intercellular carbon dioxide concentration(Ci) decreased significantly (Figure 2A–G), while water use efficiency (WUE) significantly increased at 18 d of drought stress (Figure 2H). Results showed that total Chl, Chl a, Chl b, Fv/Fm, and PIABS in the drought-stressed plants decreased significantly by 54%, 65%, 30%, 38%, or 71% when compared to that in the control after 18 days of drought stress, respectively. In contrast to control, the photosynthetic parameters Pn, Tr, Gs, and Ci in the drought-stressed plants also demonstrated a significant decline of 82%, 87%, 90%, or 46%, respectively, but WUE increased significantly by 32% after 18 days of drought stress. Exogenous Spm pretreatment significantly mitigated the drought-induced decreases in total Chl, Chl a, Chl b, Fv/Fm, PI_ABS_, Pn, Tr, Gs, and Ci, and also further increased WUE in leaves of creeping bentgrass under drought stress (Figure 2).

### 2.2. Effects of Spm Pretreatment on PAs Metabolism

To examine whether exogenous application of Spm affected endogenous PAs levels and metabolism under drought stress, endogenous PAs content and key enzymes involved in PAs metabolism were detected. As shown in Figure 3A–D, drought stress significantly induced the accumulation of Put, Spd, Spm, and total PAs. The Spm pretreatment also significantly increased endogenous Put, Spd, Spm, and total PAs contents in leaves of creeping bentgrass under normal condition and drought stress. The Put, Spd, Spm, or PAs content in leaves of plants pretreated with the Spm increased by 42%, 77%, 122%, or 48% than that in leaves of plants without the Spm pretreatment on the 18th d of droughts stress, respectively (Figure 3A–D). As time of drought stress went on, the ADC, ODC, and S-AMDC activities gradually increased in the “D” and “D + Spm” treatment. Exogenous Spm further significantly increased stress-induced ADC, ODC and S-adenosylmethionine decarboxylase (S-AMDC) activities in leaves (Figure 3E–G). Under normal condition, there were no significant differences in polyamine oxidase (PAO) activity and DAO activity between Spm-treated and untreated plants (Figure 3H,I). Drought stress caused significant increase in PAO activity and DAO activity; however, “D + Spm” treatment showed significantly lower PAO activity and higher DAO activity than the “D” treatment during drought stress (Figure 3H,I).

### 2.3. Effects of Spm Pretreatment on GABA Metabolism

To examine whether exogenous application of Spm affected GABA metabolism under drought stress, endogenous GABA content and key enzymes involved in GABA metabolism were detected. The GABA and glutamate (Glu) contents in plants increased significantly under drought stress, and the Spm-pretreated plants exhibited significantly higher GABA and Glu contents than plants without Spm application during drought stress (Figure 4A,B). The GABA aminotransferase (GABA-T) activity significantly reduced in response to drought stress, but Spm pretreatment significantly alleviated the decrease in its activity under drought stress (Figure 4C). The glutamate decarboxylase (GAD) activity and α-ketoglutarase(α-KGDH) activity were gradually increased with the extension of drought time (Figure 4D,E). The GAD activity of Spm-pretreated plants was significantly higher than that of non-pretreated plants during drought stress (Figure 4D). In addition, the activity of α-KGDH in Spm-pretreated plants was 34% or 33% higher than that in non-pretreated plants on 12th and 18th d of drought stress, respectively (Figure 4E).

### 2.4. Effects of Spm Pretreatment on N Metabolism

To examine whether exogenous application of Spm affected N metabolism under drought stress, different forms of N content and key enzymes involved in N metabolism were detected. Drought stress significantly improved the accumulation of NH_4_^+^ in leaves. Under drought stress, the content of NH_4_^+^ in plants with Spm pretreatment was 32% or 16% lower than that of untreated plants on the 12th or 18th day of drought stress, respectively (Figure 5A). The content of nitrite nitrogen (NO_2_^−^) decreased significantly in plants without the Spm pretreatment, however no significant change was observed in plants with the Spm pretreatment on the 12th d of drought stress. On the 18th day of drought stress, the content of NO_2_^−^ in Spm-pretreated plants was significantly higher than that in non-pretreated plants (Figure 5B). The NR activity was significantly inhibited under drought stress, and its activity in plants without the Spm pretreatment decreased by 36% and 62% at 12 and 18 days of drought stress, respectively, as compared to control, but Spm pretreatment significantly alleviated the decrease in NR activity induced by drought stress (Figure 5C). Drought stress significantly increased nitrite reductase (NiR) activity and Spm pretreatment significantly inhibited the increase in NiR activity as shown in Figure 5D. In addition, the Spm pretreatment could significantly alleviate the decrease in GS and GOGAT activities during 18 days of drought stress (Figure 5E,F). The GDH activity of the plants pretreated with Spm increased by 23% than that of the plant without Spm pretreatment on 18th d of drought stress (Figure 5G).

### 2.5. Effects of Spm Pretreatment on Pro Metabolism

To examine whether exogenous application of Spm affected Pro metabolism under drought stress, Pro content and key enzymes involved in Pro metabolism were detected. The content of free Pro in Spm-pretreated and non-pretreated plants increased sharply during drought stress. However, there was no significant difference in the Pro content between the Spm-pretreated and non-pretreated plants under normal or drought condition (Figure 6A). Drought stress significantly activated ornithine aminotransferase (OAT) and P5CS activities in both Spm-pretreated and non-pretreated plants (Figure 6B,C). There was no significant difference in OAT activity between the Spm-pretreated and non-pretreated plants under normal and drought conditions (Figure 6B), whereas the activity of P5CS in the Spm-pretreated plants was significantly higher than that in non-pretreated plants under drought stress (Figure 6C). The proline dehydrogenase (ProDH) activity decreased significantly in response to drought stress. On the 18th day of drought stress, the ProDH activity of the Spm-pretreated plants was significantly higher than that of non-pretreated plants (Figure 6D). Figure 7 showed the response of key metabolic pathways to drought stress and exogenous Spm pretreatment in leaves.

## 3. Discussion

Photosynthesis is an important fundamental physiological process that is responsible for the growth and biomass in higher plants and also one of the most sensitive physiological processes to environmental stress [31]. Previous studies have showed that drought stress induced water loss, photosynthetic inhibition, and the damage to membrane system in creeping bentgrass [32,33,34]. Gradual declines in leaf RWC, cell membrane stability, Chl content, Fv/Fm, PI_ABS_, and Pn in creeping bentgrass were observed due to a prolonged drought stress; however, the Spm pretreatment effectively alleviated these adverse effects caused by drought (Figure 1 and Figure 2). Chl is the main photosynthetic pigment that plays vital role in receiving and converting light energy in plants [35]. Drought stress significantly reduced the Chl a, Chl b, and total Chl content; however, the decrease in Chl a was greater than that of Chl b in leaves of creeping bentgrass. It has been observed that Chl a was more susceptible to stress than Chl b [36]. During Chl degradation, a part of Chl b can be compensated by products of Chl a degradation [37]. Relevant studies have found that PAs could protect the structural and functional integrity of chloroplasts, thereby slowing down the degradation rate of photosynthetic pigments under abiotic stress [16,38,39], which was consistent with our findings.

Stomatal movement is one of the most key regulatory mechanisms in plants affecting Tr, photosynthesis, and WUE under drought stress [40]. During drought stress, a gradual decline in Pn was accompanied by gradual decreases in Gs and Ci, indicating that the cause of the decline in photosynthetic rate in both of the Spm- and non-pretreated creeping bentgrasses was mainly affected by stomatal limitation. Plants mainly reduce Tr by restricting stomatal opening, which is conducive to reduction of water loss in leaves in response to drought stress. However, stomatal closure also diminishes gas exchange leading to the limitation of CO_2_ supply for photosynthesis. So, the homeostasis between Tr and Pn was quite important. Under drought stress, the Spm application could effectively alleviate the decrease in WUE of creeping bentgrass by maintaining significantly higher Tr and Pn in favor of the adaptation to water deficit (Figure 2). Chl fluorescence can reflect the absorption, transmission, and conversion of light energy by leaves and is used as an indicator to quantitatively analyze and compare the physiological state of plants under different environmental conditions [41]. Fv/Fm reflects the maximum photochemical quantum efficiency of photosystem II (PSII), and the photosynthetic performance index PI_ABS_ indicates the state of the plant photosynthetic system, which is more sensitive to abiotic stresses than Fv/Fm [42,43,44]. Creeping bentgrass pretreated with Spm had higher Fv/Fm and PI_ABS_ than those without Spm treatment under drought stress, which indicated that the Spm improved drought tolerance of creeping bentgrass associated with maintenance of better PSII (Figure 2).

PAs concentration varies with the growth stage and the external environment in plants. Drought leads to the higher accumulation of PAs content in leaves, which is an important strategy for plants to adapt to water-limited conditions [45]. In this current study, the Put, Spd, Spm, and total PAs content increased significantly under drought stress, and exogenous application of Spm further enhanced their accumulation in creeping bentgrass subjected to drought stress (Figure 3). These results indicated that elevated PAs levels promoted drought tolerance in creeping bentgrass. Similar results have been found in creeping bentgrass or other plant species [46,47]. In addition to PAs accumulation, PAs biosynthesis and catabolism also play pivotal roles in regulating tolerance to abiotic stresses [19,48]. ADC and ODC are two key enzymes for the biosynthesis of Put, and S-AMDC is involved in the biosynthesis of Spd and Spm. Drought stress enhanced the activities of ADC, ODC, and S-AMDC, indicating that PAs synthesis was significantly affected by these pathways. An earlier study showed that the methylglyoxal-bis-guanylhydrazone (MGBG), an inhibitor of S-AMDC activity, decreased the contents of Spm and Spd resulting in reduced drought tolerance of wheat seedlings [49]. The Spm application further improved drought-induced increases in activities of ADC, ODC, and S-AMDC (Figure 3). It could be concluded that increases in Put, Spd, and Spm contents in Spm-pretreated plants under drought stress was not only due to the direct absorption of Spm, but also due to the significant enhancement of ADC, ODC, and S-AMDC activities to promote their biosynthesis. Interestingly, drought stress also significantly enhanced activities of DAO and PAO that are two catabolic enzymes of PAs metabolism through oxidizing Put, Spd, and Spm to form GABA [50]. However, the DAO activity in Spm-pretreated creeping bentgrass was significantly higher, but PAO activity was significantly lower when compared to that in untreated plants under drought stress. This might be associated with the maintenance of higher Spd and Spm content. Previous studies found that the accumulation of Spd and Spm were more important than the Put when plants suffered from water deficit [46,51].

Apart from the catabolism of Spd and Spm, the GAD catalyzed GABA synthesis by using Glu as a substrate is an important pathway of GABA biosynthesis in plants [52]. As a non-protein amino acid, the GABA is associated with the carbon and N metabolism and has also been identified as a signaling molecule regulating growth and stress tolerance of plants [53,54]. Elevated GABA accumulation by exogenous GABA application could significantly enhance drought tolerance of creeping bentgrass [30]. The GABA branch can provide nicotinamide adenine dinucleotide (NADH) and succinate to feed the tricarboxylic acid (TCA) cycle [52]. This process is then of primary importance because enhanced TCA cycle could provide more energy for growth and stress defense under unfavorable environmental conditions [55]. Current findings showed that the Spm application further significantly improved the drought-induced increases in Glu content, GABA content, and GAD activity (Figure 4). These findings indicated that the Spm induced GABA accumulation mainly associated with the activation of GAD and DAO in leaves of creeping bentgrass. In addition, GABA-T activity declined, but α-KGDH activity increased gradually in creeping bentgrass during drought stress. The GABA-T catalyzes the transformation between GABA and succinic semialdehyde. The α-KGDH is a key enzyme in the tricarboxylic acid cycle and its function is to decompose α-KGDH in plants [56]. Under drought stress, the Spm could promote the TCA cycle through activating α-KGDH and GABA-T activities contributing towards enhanced drought tolerance. The Spm-pretreated creeping bentgrass exhibited significantly higher α-KGDH and GABA-T activities than untreated plants during drought stress (Figure 4). Our earlier study also demonstrated that the supply of GABA could maintain higher intermediates of TCA cycle in leaves of creeping bentgrass under heat stress, thereby alleviating heat-induced metabolic imbalance and damage [57].

The Glu plays a central metabolic role in N assimilation pathways and can be used as a source of N for the biosynthesis of other amino acids such as Pro and GABA [58,59,60]. N metabolism is one of the most fundamental metabolic processes, affecting growth status, yield, and stress tolerance in plants [61]. Drought stress reduced the uptake and assimilation of main inorganic nitrogenous compounds including NH_4_^+^ and nitrate nitrogen (NO_3_^−^) by roots [62]. NR is the key synthase of NO_2_^−^ while NiR is the main enzyme to produce NH_4_^+^ [63]. It has been reported that drought or salt stress reduced NR activity and NO_2_^−^ content in various plant species [22,64,65]. Similar results were also found in our present study. However, Spm pretreatment significantly alleviated declines in NR activity and NO_2_^−^ content in leaves of creeping bentgrass under drought stress (Figure 5). Interestingly, drought stress resulted in immense accumulation of NH_4_^+^ and significant increase in NiR activity in creeping bentgrass, and the Spm application alleviated their increases induced by drought stress. NH_4_^+^ can be produced by NiR catalyzing NO_2_^−^ and many other pathways in plants [66]. Reduced accumulation of NH_4_^+^ in plant tissues is considered to be an important ability to resist drought stress, since the overproduction of NH_4_^+^ cause toxicity in cells [63]. GS, GOGAT, and GDH are three key rate-limiting enzymes in the process of NH_4_^+^ assimilation and catalyze the biosynthesis of Glu by using NH_4_^+^ in plants [67]. Drought stress improved NiR activity and decreased GS and GOGAT activities, which might result in the overaccumulation of NH_4_^+^ in creeping bentgrass. The Spm pretreatment could significantly alleviate drought-induced decrease in GS and GOGAT activities and also maintained significantly higher GDH activity, which indicated the Spm-mitigated inhibition of NH_4_^+^ assimilation possibly associated with the mediation of GDH and GS/GOGAT pathways (Figure 5). In addition, the improvement in GDH activity induced by the Spm contributed to the accumulation of Glu in creeping bentgrass under drought stress. The GDH can transform glutamic acid into α-ketoglutarate associated with the TCA cycle in plants. The improvement of N assimilation and alleviation of ammonia toxicity could be the important regulatory mechanisms in relation to the Spm-regulated drought tolerance in creeping bentgrass.

As a stress-inducible amino acid, the function of Pro in response to drought stress has been widely elucidated in plants. Its accumulation and metabolism are involved in osmotic adjustment and reactive oxygen species (ROS) scavenging, thus regulating the stability of intracellular structure and membrane system under drought stress [68]. Pro content is also often used as an important index to estimate the severity of stress injury in plants [69]. In this present study, Pro content in the leaves of creeping bentgrass increased rapidly during drought stress (Figure 6A), but the Pro content in Spm-pretreated plants was not significantly different from that in the untreated plants. Pro biosynthesis is dependent on Glu pathway and ornithine pathway, and OAT and P5CS are the key enzymes in these two pathways. ProDH is a chief enzyme in Pro catabolism pathway and it catalyzes the degradation of Pro to produce Glu in plants [70]. Drought stress significantly improved OAT and P5CS activities, but significantly inhibited the ProDH activity in creeping bentgrass with and without the Spm application under drought stress (Figure 6). This indicated that increase in Pro accumulation could be an important adaptive response in creeping bentgrass under drought stress. Although there was no significant difference in OAT activity between Spm-pretreated and non-pretreated plants under drought stress (Figure 6B), but the Spm-pretreated creeping bentgrass maintained significantly higher P5CS and ProDH activities than untreated plants under drought stress (Figure 6C,D). Similar results were found in our previous studies which demonstrated that exogenous Spm improved drought tolerance of white clover associated with higher Pro metabolism by activating P5CS and ProDH activities [71]. Maintenance of higher Pro metabolism instead of more Pro accumulation could be one of the main regulatory mechanisms for the Spm-induced drought tolerance in plants, which will be beneficial for the improved Glu cycle and metabolic homeostasis of amino acids under water deficient conditions.

## 4. Conclusions

Drought tolerance of creeping bentgrass could be significantly improved by the Spm pretreatment, as demonstrated by the maintenance of less Chl loss and higher photosynthesis, WUE, and cell membrane stability. The Spm pretreatment further increased drought-induced accumulation of endogenous PAs, Put, Spd, and Spm, and also enhanced PAs metabolism through improving ADC, ODC, and S-AMDC activities during drought stress. The Spm application not only significantly enhanced GABA shunt through improving endogenous GABA content, Glu content, and activities of GAD and α-KGDH, but also alleviated decline in NO_2_^−^ content, NR, GS, GOGAT, and GABA-T activities under drought stress. The Spm-pretreated creeping bentgrass exhibited significantly lower ammonia N content and NiR activity as well as higher GDH activity than non-pretreated plants in response to drought stress. These results indicated a beneficial function of the Spm in regulating GABA and N metabolism contributing towards better maintenance of the TCA cycle in creeping bentgrass. In addition, the Spm-enhanced Pro metabolism rather than more Pro accumulation could be a key regulatory mechanism for drought tolerance in creeping bentgrass (Figure 7). Current findings will provide a comprehensive understanding of PAs interaction with other metabolic pathways regulating drought tolerance in grass species.

## 5. Materials and Methods

### 5.1. Plant Materials and Treatments

Seeds of creeping bentgrass cv. ‘PA-4’ were purchased from the Tee-2-Green company (Hubbard, OR, USA) and sown in plastic containers (24 cm length, 18 cm width, 9 cm depth) with sterile quartz sands in a controlled growth chamber (photoperiod cycle of 10/14 h light/dark, 21/18 °C day/night, 65% relative humidity, and 700 μmol m^−2^·s^−1^ PAR). After being germinated in distilled water for 7 days, the seedlings were cultured in Hoagland’s nutrient solution [72] for 14 days. Plants were then moved carefully from quartz sands and cultivated in the Hoagland’s nutrient solution for 7 days of hydroponics. These plants were pretreated with or without 100 μmol/L Spm in roots for 3 days [73,74]. The Spm-pretreated and untreated plants were subjected to normal conditions or drought stress induced by 18% polyethylene glycol (PEG) 6000 (−0.5 MPa) that was dissolved in Hoagland’s nutrient solution for 18 days. The experimental design was completely random. Fresh nutrient solution or PEG solution was applied every day. Each treatment had four independent biological replicates and the experiment was reproduced four times, independently. Leaf samples were taken at 0, 12, and 18 days.

### 5.2. Determination of Leaf Water Status, Cell Membrane Stability, and Photosynthesis

The leaf RWC was determined according to the method of Barrs and Weatherley [75]. The EL was measured based on the method of Blum and ebercon [76]. The total Chl, Chl a, and Chl b content were estimated by following the procedure of Arnon [77]. A portable Chl fluorescence system (Pocket PEA, Hansatech, UK) was used for Fv/Fm and PI_ABS_. Photosynthetic parameters including Pn, Tr, Ci, Gs, and WUE were estimated by using a portable photosynthetic system (CIRAS-3, PP system, Amesbury, MA, USA).

### 5.3. Determination of PAs Metabolism

Endogenous PAs content was determined according to the method of liquid chromatography [46]. 0.1 g of leaf samples were collected and ground with 1 mL precooled 5% perchloric acid (*v*/*v*). The mixture was centrifuged (12,000× *g*, 4 °C) for 15 min twice to get the supernatant. The reaction mixture comprising of 400 μL supernatant, 7 μL benzoyl chloride, and 1 mL NaOH (2 mol/L) was shaken well for 20 s and kept at 37 °C in a water bath for 20 min. After this, 2 mL saturated NaCl solution was added, gently mixed, and 2 mL ether (AR) was supplemented. The mixture was then centrifuged for 2 min (12,000× *g*, 4 °C). The 1 mL of the ether phase was taken out and put in a 2 mL centrifuge tube until evaporating to dryness at 4 °C. The 1 mL of methanol was added, shaken up evenly, and filtered through 0.25 μm organic phase filter membrane. The 20 μL of extractives was used for the HPLC. The HPLC column was Novapak C18 (Waters Associates, Milford, MA, USA), and the mobile phase was 36% water and 64% methanol. The column temperature, the flow rate, and the wavelength were 25 °C, 0.7 mL/min, and 230 nm, respectively. Chromatographic-grade Put (Merck Co., Kenilworth, NI, USA, 51799-100MG), Spd (Merck Co., S2626-1G), and Spm (Merck Co., S3256-1G) were used for internal standards. For the activities of PAs-metabolic enzymes, ADC, ODC, or S-AMDC activity were measured by using kits purchased from MLB Good ELISA Kit producers, Shanghai, China. The activity of PAO (Art. No. G0135F) or DAO (Art. No. G0134F) was measured using a kit obtained from Suzhou Grace Biotechnology Co., Ltd., Suzhou, China.

### 5.4. Determination of Pro Metabolism

The content of free Pro was estimated by ninhydrin colorimetry method [78]. Leaf samples (0.1 g) were immerged in 3 mL (3%) sulfosalicylic acid solution in a 15 mL of test tube and placed in boiling water bath for 20 min followed by cooling in an ice bath to room temperature. After being centrifuged, the supernatant was collected, and 1mL of supernatant was mixed with 1 mL glacial acetic acid and 1.5 mL chromogenic solution. The reaction solution was kept in a boiling water bath for 40 min, and cooled in an ice bath to room temperature. The 2.5 mL toluene was added and shaken swiftly for 10 min. Later, the mixture was kept in dark conditions for 10 min, and the absorbance value of toluene layer was measured at 520 nm. OAT activity, P5CS activity, and ProDH activity were determined by following Sánchez method [79], respectively.

### 5.5. Determination of N Metabolism and GABA Metabolism

To determine the GS activity, 0.1 g leaf sample was added in 1 mL buffer (pH 8.0, 0.05 mol/L Tris HCl) containing 2 mmol/L Mg^2+^, 2 mmol/L DTT, and 0.4 mol/L sucrose, and ground in ice bath, followed by centrifugation for 20 min (15,000 × r, 4 °C). The 0.35 mL of supernatant was collected and mixed with 0.8 mL of reaction mixture (80 mmol/L hydroxylamine hydrochloride, 0.1 mmol/L Tris HCl buffer, 80 mmol/L Mg^2+^, 20 mmol/L sodium glutamate, 20 mmol/L cysteine, and 2 mmol/L EDTA, pH 7.4) and 0.35 mL of (ATP) solution (40 mmol/L). The mixture was kept at 37 °C for 30 min, and then 5 mL of chromogenic solutions (0.2 mol/L TCA, 0.37 mol/L, and FeCl_3_ 0.6 mol/L) were added into the reaction mixture. After being centrifuged for 10 min (5000× *g*, 4 °C), the supernatant was collected and absorbance value of the supernatant was detected at 540 nm. The assay method of nitrite nitrogen (NO_2_^−^) content, ammonium nitrogen (NH_4_^+^) content, GOGAT activity, or GDH activity in details has been provided in our previous study [30]. The NiR activity (Art. No. G0408W) and NR activity were measured using a kit purchased from Suzhou Grace Biotechnology Co., Ltd., Suzhou, China. The GABA content (Art. No. G1106F), Glu content (Art. No. G0427F), GAD activity (Art. No. G1102W), GABA transaminase (GABA-T) activity (Art. No. G1103W), andα-KGDH activity (Art. No. G0840F) were measured by using a kit purchased from Suzhou Grace Biotechnology Co., Ltd., Suzhou, China.

### 5.6. Statistical Analysis

The experiment design was a split-plot design with water status as the main plot and Spm treatment as the subplot. Statistical analysis was conducted using Statistix 8.1 (version, 8.1. Statistix, Tallahassee, FL, USA). Significant differences among all treatments were measured by using ANOVA (one way) in combination with LSD (least significant difference) test. The significance of differences was evaluated at the 5 percent probability level (*p* < 0.05).

## Figures and Tables

**Figure 1 ijms-23-02779-f001:**
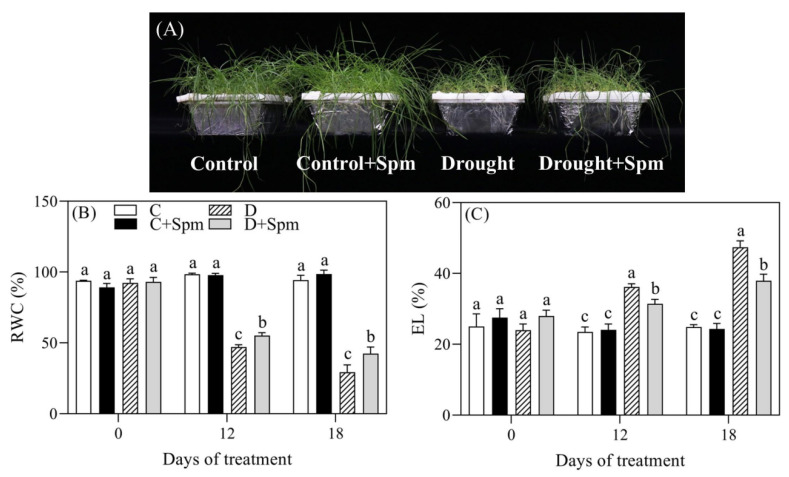
Effects of Spm pretreatment on (**A**) phenotypic change, (**B**) relative water content (RWC), and (**C**) electrolyte leakage (EL) in leaves of creeping bentgrass under normal condition and drought stress. Vertical bars indicate ± SE of mean (*n* = 4). The “*n* = 4” indicates four independent biological replicates. Different letters above columns indicate significant difference at a particular day based on the LSD (*p* ≤ 0.05). C, well-watered control; C + Spm, well-watered control pretreated with Spm; D, drought stress; D + Spm, drought-stressed plants pretreated with Spm.

**Figure 2 ijms-23-02779-f002:**
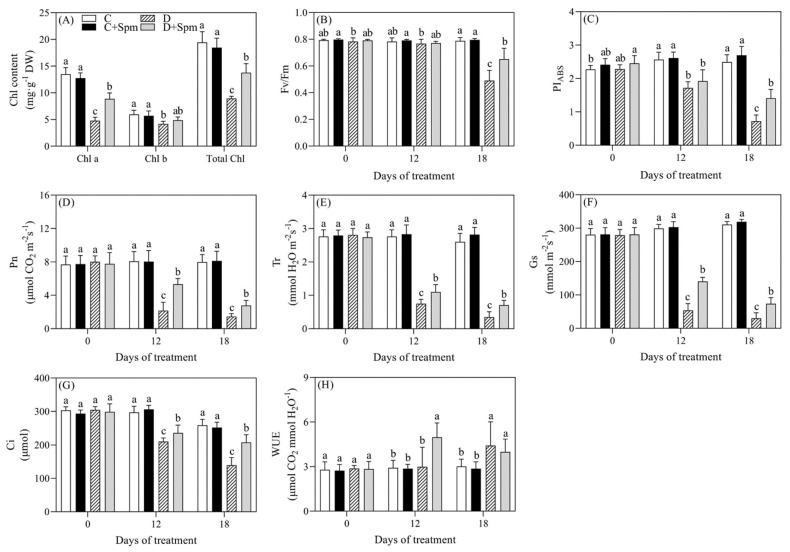
Effects of Spm pretreatment on (**A**) total chlorophyll content (Total Chl), chlorophyll a (Chl a) and chlorophyll b (Chl b) under drought stress for 18 days; Effects of Spm pretreatment on (**B**) photochemical efficiency (Fv/Fm), (**C**) performance index on absorption basis (PI_ABS_), (**D**) net photosynthetic rate (Pn), (**E**) transpiration rate(Tr), (**F**) stomatal conductance(Gs), (**G**) intercellular carbon dioxide concentration(Ci), and (**H**) water use efficiency (WUE) in leaves of creeping bentgrass under normal condition and drought stress. Vertical bars indicate ± SE of mean (*n* = 4). The “*n* = 4” indicates four independent biological replicates. Different letters above columns indicate significant difference at a particular day based on the LSD (*p* ≤ 0.05). C, well-watered control; C + Spm, well-watered control pretreated with Spm; D, drought stress; D + Spm, drought-stressed plants pretreated with Spm.

**Figure 3 ijms-23-02779-f003:**
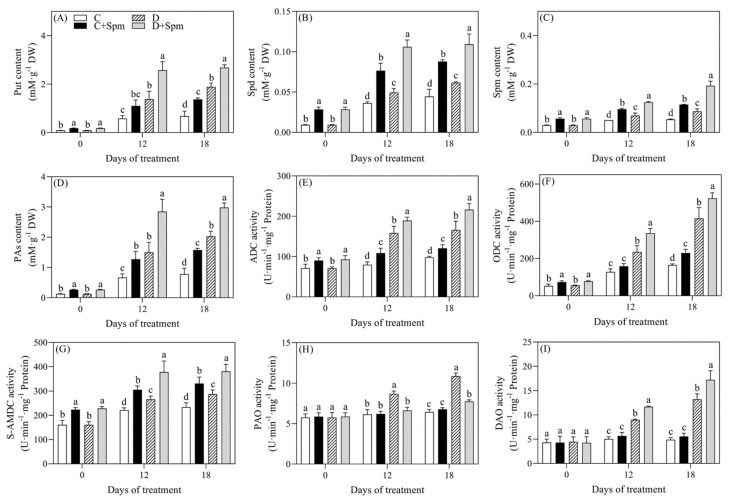
Effects of Spm pretreatment on (**A**) putrescine (Put) content, (**B**) spermidine (Spd) content, (**C**) spermine (Spm) content, (**D**) polyamines (PAs) content, (**E**) arginine decarboxylase (ADC) activity, (**F**) ornithine decarboxylase (ODC) activity, (**G**) S-adenosylmethionine decarboxylase (S-AMDC) activity, (**H**) polyamine oxidase (PAO) activity, and (**I**) diamine oxidase (DAO) activity in leaves of creeping bentgrass under normal condition and drought stress. Vertical bars indicate ± SE of mean (*n* = 4). The “*n* = 4” indicates four independent biological replicates. Different letters above columns indicate significant difference at a particular day based on the LSD (*p* ≤ 0.05). C, well-watered control; C + Spm, well-watered control pretreated with Spm; D, drought stress; D + Spm, drought-stressed plants pretreated with Spm.

**Figure 4 ijms-23-02779-f004:**
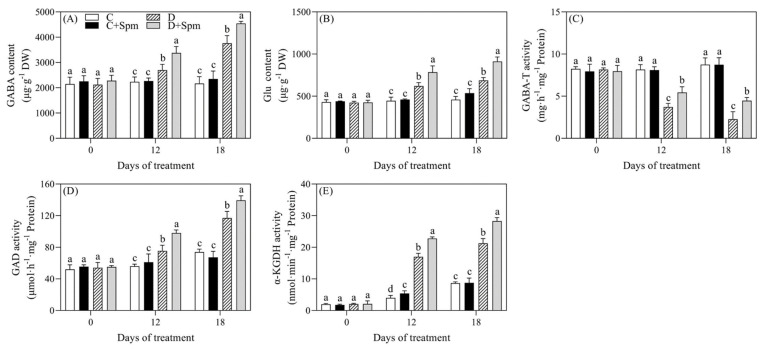
Effects of Spm pretreatment on (**A**) γ-aminobutyric acid (GABA) content, (**B**) glutamate (Glu) content, (**C**) GABA aminotransferase (GABA-T) activity, (**D**) glutamate decarboxylase (GAD) activity and (**E**) α-ketoglutarase(α-KGDH) activity in leaves of creeping bentgrass under normal condition and drought stress. Vertical bars indicate ± SE of mean (*n* = 4). The “*n* = 4” indicates four independent biological replicates. Different letters above columns indicate significant difference at a particular day based on the LSD (*p* ≤ 0.05). C, well-watered control; C + Spm, well-watered control pretreated with Spm; D, drought stress; D + Spm, drought-stressed plants pretreated with Spm.

**Figure 5 ijms-23-02779-f005:**
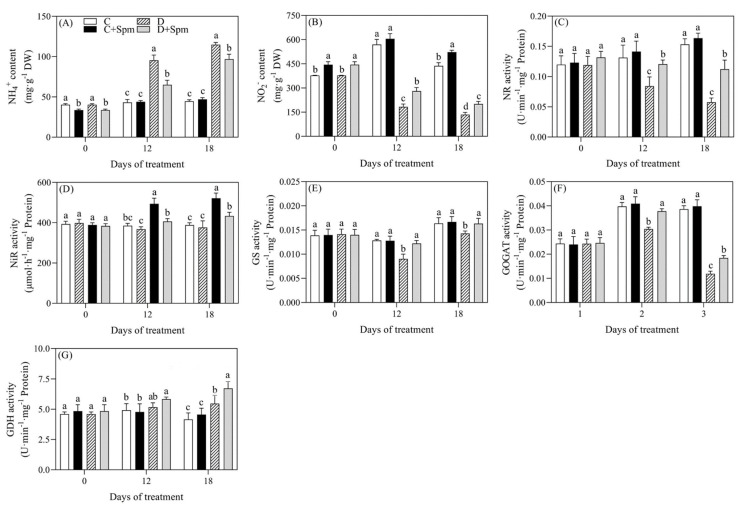
Effects of Spm pretreatment on (**A**) ammonium nitrogen (NH_4_^+^) content, (**B**) nitrite nitrogen (NO_2_^−^) content, (**C**) nitrate reductase (NR) activity, (**D**) nitrite reductase (NiR) activity, (**E**) glutamine synthetase (GS) activity, (**F**) glutamate synthetase (GOGAT) activity and (**G**) glutamate dehydrogenase (GDH) activity in leaves of creeping bentgrass under normal condition and drought stress. Vertical bars indicate ± SE of mean (*n* = 4). The “*n* = 4” indicates four independent biological replicates. Different letters above columns indicate significant difference at a particular day based on the LSD (*p* ≤ 0.05). C, well-watered control; C + Spm, well-watered control pretreated with Spm; D, drought stress; D + Spm, drought-stressed plants pretreated with Spm.

**Figure 6 ijms-23-02779-f006:**
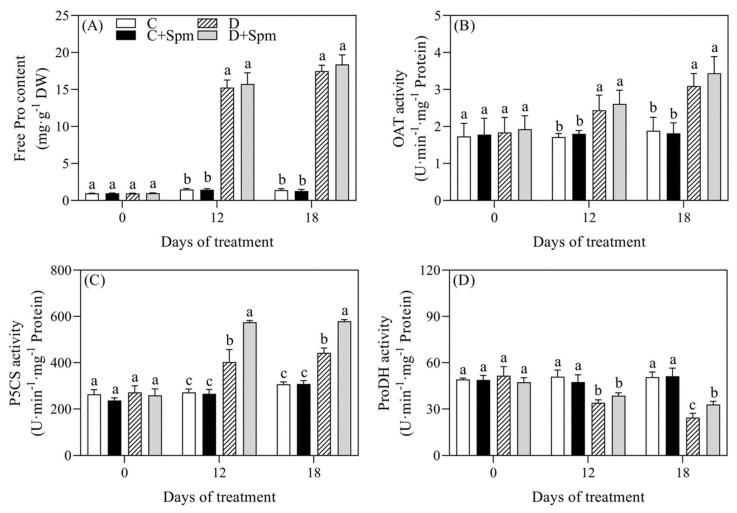
Effects of Spm pretreatment on (**A**) free proline (Pro) content, (**B**) ornithine aminotransferase (OAT) activity, (**C**) Δ’- pyrroline-5-carboxylic acid synthetase (P5CS) activity and (**D**) proline dehydrogenase (ProDH) activity in leaves of creeping bentgrass under normal condition and drought stress. Vertical bars indicate ± SE of mean (*n* = 4). The “*n* = 4” indicates four independent biological replicates. Different letters above columns indicate significant difference at a particular day based on the LSD (*p* ≤ 0.05). C, well-watered control; C + Spm, well-watered control pretreated with Spm; D, drought stress; D + Spm, drought-stressed plants pretreated with Spm.

**Figure 7 ijms-23-02779-f007:**
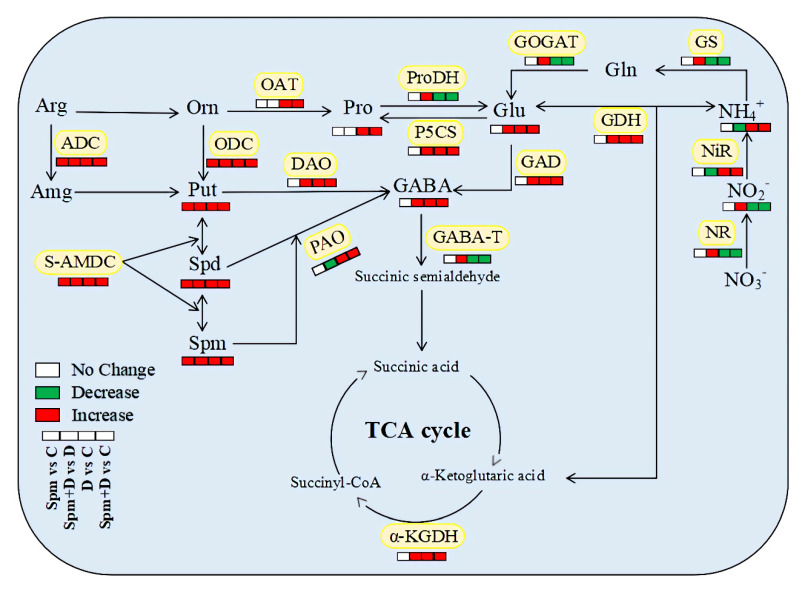
Response of key metabolic pathways in leaves of creeping bentgrass with or without the spermine pretreatment after 18 days of drought stress.

## Data Availability

Not applicable.

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
