# Peer review of "Polyamines Metabolism Interacts with γ-Aminobutyric Acid, Proline and Nitrogen Metabolisms to Affect Drought Tolerance of Creeping Bentgrass"

_ijms, 2022, doi:10.3390/ijms23052779_

Round 1
Reviewer 1 Report
The presented for review work of Tan, M.; Hassan, M.J.; Peng, Y.; Feng, G.; Huang, L.; Liu, L.; Liu, W.; Han, L.; Li, Z. Polyamines metabolism interacts with γ-amino- butyric acid, proline and nitrogen metabolisms to affect drought tolerance of creeping bentgrass",
is a continuation and extension of the authors' previous research Tang et al. 2020 Int. J. Mol. Sci. 2020, 21(20), 7460; https://doi.org/10.3390/ijms21207460.
In the present study, the authors explained the potential effects of PAs metabolism on Pro metabolism, N absorption and assimilation, and GABA metabolism under drought stress.
Their results are consistent and well presented in numerous charts.
The results seem to be unambiguous and statistically significant, although it is not known what statistical test was used?
I have not found such information in the descriptions of figures nor in the chapter describing materials and methods, which is carelessly prepared and requires correction.
Paragraph 5.6. mentioning the software used for statistical analyses, but unfortunately, it does not give information about the test used.
Moreover, the title of the paragraph is the copied title of the previous paragraph 5.5.
The chapter Results does not raise any objections, except for the last sentence.
The authors present the highly significant Figure 7.
This figure summarizes all the results obtained. It is the quintessence of the presented research. However, it is described extremely laconically because it is only in one sentence
"Fig. 7 showed the response of key metabolic pathways to drought stress and exogenous Spm pretreatment in leaves".
Such a way of describing the proposed scheme is unsatisfactory.
An explanation of the scheme presented in figure 7 can be found in the paragraph containing the discussion, but the authors have nowhere included an appropriate reference to this figure.
I believe that this is an essential part of the work and requires much more attention, and some rearrangement of the manuscript organisation should have to be performed.
English needs to be corrected.
The authors use laboratory jargon in some places.
For example, the first sentence from "Section 5.5. Determination of N metabolism and GABA metabolism".
"For the GS activity, 0.1 g leaf sample was added in 1 mL extract (pH 8.0, 0.05 mol / L Tris HCl) containing 2 mmol / L Mg2 +, 2 mmol / L DTT, and 0.4 mol / L sucrose, and ground in ice bath, followed by centrifugation for 20 min (15,000 y, 4 ° C) ".
Shouldn't the above sentence begin with the words: To determine the GS activity ...?
Also, I propose that the word extract should be replaced with buffer?
Similarly, in the first sentence of paragraph "2.5. Effects of Spm pretreatment on Pro metabolism".
Perhaps the sentence should start as follows?: The content of free Pro in Spm-pretreated and non-pretreated plants increased sharply during drought stress.
Many abbreviations in the manuscript are being clarified only in the figure captions and in the Materials and Methods section, situated at the end of the manuscript. Lack of explanations in the place where the abbreviation appears for the first time makes the perception of the text much more difficult.
The abbreviation of MGBG is missing. Please enter a full name
One final small note, there are no spaces in front of the references in the sentence.
"These results indicated that elevated PAs levels promoted drought tolerance in creeping bentgrass. Similar results have been found in creeping bentgrass or other plant species [35.53]".
Author Response
In the present study, the authors explained the potential effects of PAs metabolism on Pro metabolism, N absorption and assimilation, and GABA metabolism under drought stress. Their results are consistent and well presented in numerous charts.
Reply: Thank you very much for your professional and valuable review. We have carefully revised the manuscript according to suggestions.
- The results seem to be unambiguous and statistically significant, although it is not known what statistical test was used? I have not found such information in the descriptions of figures nor in the chapter describing materials and methods, which is carelessly prepared and requires correction.
Reply: Thanks. The statistical analysis has been mentioned in Materials and Methods section “Paragraph 5.6” (line 498-502).
- Paragraph 5.6. mentioning the software used for statistical analyses, but unfortunately, it does not give information about the test used.
Reply: Thank you for your careful review. The software used for statistical analyses has been added in the “5.6 Statistical Analysis” section (line 498-502)
- Moreover, the title of the paragraph is the copied title of the previous paragraph 5.5.
Reply: Thanks. It was a typing error. The title of 5.6 has been revised to “5.6. Statistical Analyses” (line 494).
- An explanation of the scheme presented in figure 7 can be found in the paragraph containing the discussion, but the authors have nowhere included an appropriate reference to this figure. I believe that this is an essential part of the work and requires much more attention, and some rearrangement of the manuscript organisation should have to be performed.
Reply: Thank you very much for your valuable suggestion The figure 7 has been cited in the Results section (line 242-244) and the Discussion section (line 412) according to suggestion.
- "For the GS activity,0.1 g leaf sample was added in 1 mL extract (pH 8.0, 0.05 mol / L Tris HCl) containing 2 mmol / L Mg2 +, 2 mmol / L DTT, and 0.4 mol / L sucrose, and ground in ice bath, followed by centrifugation for 20 min (15,000 y, 4 ° C) ". Shouldn't the above sentence begin with the words: To determine the GS activity...? Also, I propose that the word “extract” should be replaced with “buffer”?
Reply: Thank you very much for your good suggestions. We have replaced “For the GS activity” with “To determine the GS activity” and revised the word “extract” to “buffer” (line 475).
- Similarly, in the first sentence of paragraph "2.5. Effects of Spm pretreatment on Pro metabolism". Perhaps the sentence should start as follows ? : The content of free Pro in Spm-pretreated and non-pretreated plants increased sharply during drought stress.
Reply: Thanks. This sentence has been modified as suggested (line 231).
- Many abbreviations in the manuscript are being clarified only in the figure captions and in the Materials and Methods section, situated at the end of the manuscript. Lack of explanations in the place where the abbreviation appears for the first time makes the perception of the text much more difficult.
Reply: Thank you very much for your careful review. Full names of all abbreviations have been added when these abbreviations appeared for the first time in the manuscript (line 67, line 84, line 89, line 117-118, line131-135, line 165-167, line 182, line 185, line 187-188, line 207, line 215, line 234-235, line 240, line 305-306, line 326-327, line 348-349, line 374, line 431-432, line 435-439, line 457-459, line 471-472, line 487, line 489-491).
- The abbreviation of MGBG is missing. Please enter a full name
Reply: The full name of MGBG has been added in revised manuscript (line 305-306).
- One final small note, there are no spaces in front of the references in the sentence. "These results indicated that elevated PAs levels promoted drought tolerance in creeping bentgrass. Similar results have been found in creeping bentgrass or other plant species [35.53]".
Reply: Thank you very much for your careful review. The space has been added between “…other plant species” and “[35.53]” (line 300).
Reviewer 2 Report
This paper presents the effect of spermine on the tolerance of creeping bentgrass to drought stress. The authors assessed the photosynthesis status, the polyamine, GABA, nitrogen and proline metabolism in creeping bentgrass plants under drought stress, with or without a pre-treatment with spermine. Their results indicate that a spermine pre-treatment effectively increases drought tolerance of creeping bentgrass.
Although the information provided by the authors in this paper is not particularly original, the results are well presented, well-structured and provide the readers with some advances in their understanding of the spermine effect on drought tolerance at the molecular level.
The study is correctly designed, few methodological details need to be added or justified and conclusions are supported by the results. However, the findings in the results section are ‘catalogued’, the experimental design is not justified, the authors do not provide information about the hypothesis behind each experiment, why they did these experiments. In addition, the authors need to justify their choice of a LSD for statistics. From their statistical analyses, there is statistical significance when comparing data that appears fairly non-significant to me when looking at the vertical bars representing standard errors in their bar plots. The authors need to provide details about the statistical test they used, why they choose this specific test (number of technical and biological replicates, normality of the data…) for the data to be assessed adequately by IJMS reviewers.
Specific comments are listed below:
- The abstract is too long; the authors should consolidate the information given in a more synthetic version. It contains experimental details that are not relevant and should be included in the result section and material & methods section (line 19-20).
_ In the results section, the authors do not explain why they decided to use 100uM/L of spermine in their experiments. Usually, a dose-dependent effect is performed first to determine the right concentration of the metabolite to use for treatments. The authors need to, either give the reference of the published papers that use a similar concentration, or present results showing creeping bentgrass phenotypic changes in response to different concentration of spermine.
_ In Figure 2, 3, 4 5 and 6 the authors need to review their statistical analyses before their results can be considered for publication.
_ The results partially support the conclusions emitted by the authors. Their conclusions are only based on the effect of spermine on the metabolism of creeping bentgrass but their study does not involve any genetic analyses. Therefore, they should be careful when drawing conclusions. For example, they mention that the spermine treatment mitigated the inhibition of ammonia assimilation through regulating GDH and GS/GOGAT pathway. This can only be claimed by using genetic approaches.
_ The authors do not provide information about the internal standards they used, for the determination of the endogenous PAs content in the Material & methods.
_ Section 5.6 of the Material and Methods do not present the determination of the N and GABA metabolism as entitled.
Author Response
Although the information provided by the authors in this paper is not particularly original, the results are well presented, well-structured and provide the readers with some advances in their understanding of the spermine effect on drought tolerance at the molecular level.
Reply: Thank you very much for your professional and valuable review. We have carefully revised the manuscript according to suggestions.
- The study is correctly designed, few methodological details need to be added or justified and conclusions are supported by the results. However, the findings in the results section are ‘catalogued’, the experimental design is not justified, the authors do not provide information about the hypothesis behind each experiment, why they did these experiments.
Reply: Thank you for you good suggestions. The experimental design has been added in the section “5.6 Statistical Analysis” (line 495-496). The information about the hypothesis behind each experiment has been added in the Results section (line 116-118, line 154-156, line 180-182, line 202-204, line 229-230).
- In addition, the authors need to justify their choice of a LSD for statistics. The authors need to provide details about the statistical test they used, why they choose this specific test (number of technical and biological replicates, normality of the data…) for the data to be assessed adequately by IJMS reviewers.
Reply: Thank you very much for your careful review. we have provided the details about the statistical test used for the significance of data in the material methods section “Paragraph 5.6” (line 498-502). We used this specific test for our experiment based on some relevant previous studies such as “DOI: 10.3390/plants9040467-Selenium and Salt Interactions in Black Gram (Vigna mungo L.): Ion Uptake, Antioxidant Defense System, and Photochemistry Efficiency”, “DOI: 10.3390/plants9111575- Effect of Cadmium Toxicity on Growth, Oxidative Damage, Antioxidant Defense System and Cadmium Accumulation in Two Sorghum Cultivars”. Four independent biological replicates for each treatment were used in our current study, which has been demonstrated in the “5.1. Plant materials and treatments” section (line 428).
Specific comments are listed below:
- The abstract is too long; the authors should consolidate the information given in a more synthetic version. It contains experimental details that are not relevant and should be included in the result section and material & methods section (line 19-20).
Reply: Thanks. The experimental details and other unnecessary informations have been deleted to reduce the length of the abstract according to suggestion (line 13-16, line 19-21).
- In the results section, the authors do not explain why they decided to use 100uM/L of spermine in their experiments. Usually, a dose-dependent effect is performed first to determine the right concentration of the metabolite to use for treatments. The authors need to, either give the reference of the published papers that use a similar concentration, or present results showing creeping bentgrass phenotypic changes in response to different concentration of spermine.
Reply: Thanks. The dose of Spermine 100 uM/L has been selected based on previous studies of our group, because this dose (100 uM/L) could significantly improve the tolerance to abiotic stress in creeping bentgrass. Relevant references have been added in this section according to suggestions (line 424, line 672-676).
- In Figure 2, 3, 4 5 and 6 the authors need to review their statistical analyses before their results can be considered for publication.
Reply: Thanks. We have carefully reviewed the statistical analyses used for all the above mentioned figures.
- The results partially support the conclusions emitted by the authors. Their conclusions are only based on the effect of spermine on the metabolism of creeping bentgrass but their study does not involve any genetic analyses. Therefore, they should be careful when drawing conclusions. For example, they mention that the spermine treatment mitigated the inhibition of ammonia assimilation through regulating GDH and GS/GOGAT pathway. This can only be claimed by using genetic approaches.
Reply: Thank you very much for your careful review. In this experiment, we focused on the physiological and metabolic regulations associated with Spm-induced drought tolerance in creeping bentgrass through analyzing various metabolites and enzymes involved in biosynthesis and catabolism of these metabolites. As you mentioned, genetic approaches could be used for analyzing changes in metabolic pathways. However, changes in metabolites and enzyme activities are also widely used to evaluate alterations in metabolic pathways.
- The authors do not provide information about the internal standards they used, for the determination of the endogenous PAs content in the Material & methods.
Reply: Thank you very much for your careful review. Internal standards we used for determination of endogenous PAs have been added in line 454-456.
- Section 5.6 of the Material and Methods do not present the determination of the N and GABA metabolism as entitled.
Reply: Thanks. It was a typing error. The title of 5.6 has been revised to “5.6. Statistical Analyses” (line 494).
Round 2
Reviewer 2 Report
Although the information provided by the authors in this paper is not particularly original, the results are well presented, well-structured and provide the readers with some advances in their understanding of the spermine effect on drought tolerance at the molecular level.
Reply: Thank you very much for your professional and valuable review. We have carefully revised the manuscript according to suggestions.
- The study is correctly designed, few methodological details need to be added or justified and conclusions are supported by the results. However, the findings in the results section are ‘catalogued’, the experimental design is not justified, the authors do not provide information about the hypothesis behind each experiment, why they did these experiments.
Reply: Thank you for you good suggestions. The experimental design has been added in the section “5.6 Statistical Analysis” (line 495-496). The information about the hypothesis behind each experiment has been added in the Results section (line 116-118, line 154-156, line 180-182, line 202-204, line 229-230).
Reviewer’s comment: The experimental design is better justified now. Minor English editing is required.
- In addition, the authors need to justify their choice of a LSD for statistics. The authors need to provide details about the statistical test they used, why they choose this specific test (number of technical and biological replicates, normality of the data…) for the data to be assessed adequately by IJMS reviewers.
Reply: Thank you very much for your careful review. we have provided the details about the statistical test used for the significance of data in the material methods section “Paragraph 5.6” (line 498-502). We used this specific test for our experiment based on some relevant previous studies such as “DOI: 10.3390/plants9040467-Selenium and Salt Interactions in Black Gram (Vigna mungo L.): Ion Uptake, Antioxidant Defense System, and Photochemistry Efficiency”, “DOI: 10.3390/plants9111575- Effect of Cadmium Toxicity on Growth, Oxidative Damage, Antioxidant Defense System and Cadmium Accumulation in Two Sorghum Cultivars”. Four independent biological replicates for each treatment were used in our current study, which has been demonstrated in the “5.1. Plant materials and treatments” section (line 428).
Reviewer’s comment: I would suggest the authors to specify the number of plants included in their biological replicates. Is n=4 plants considered as four biological replicates? Or did they use a pool of n plants per biological replicate, and they reproduce the experiment four times, independently? This must be specified to assess the statistics properly.
Specific comments are listed below:
- The abstract is too long; the authors should consolidate the information given in a more synthetic version. It contains experimental details that are not relevant and should be included in the result section and material & methods section (line 19-20).
Reply: Thanks. The experimental details and other unnecessary informations have been deleted to reduce the length of the abstract according to suggestion (line 13-16, line 19-21).
- In the results section, the authors do not explain why they decided to use 100uM/L of spermine in their experiments. Usually, a dose-dependent effect is performed first to determine the right concentration of the metabolite to use for treatments. The authors need to, either give the reference of the published papers that use a similar concentration, or present results showing creeping bentgrass phenotypic changes in response to different concentration of spermine.
Reply: Thanks. The dose of Spermine 100 uM/L has been selected based on previous studies of our group, because this dose (100 uM/L) could significantly improve the tolerance to abiotic stress in creeping bentgrass. Relevant references have been added in this section according to suggestions (line 424, line 672-676).
- In Figure 2, 3, 4 5 and 6 the authors need to review their statistical analyses before their results can be considered for publication.
Reply: Thanks. We have carefully reviewed the statistical analyses used for all the above mentioned figures.
- The results partially support the conclusions emitted by the authors. Their conclusions are only based on the effect of spermine on the metabolism of creeping bentgrass but their study does not involve any genetic analyses. Therefore, they should be careful when drawing conclusions. For example, they mention that the spermine treatment mitigated the inhibition of ammonia assimilation through regulating GDH and GS/GOGAT pathway. This can only be claimed by using genetic approaches.
Reply: Thank you very much for your careful review. In this experiment, we focused on the physiological and metabolic regulations associated with Spm-induced drought tolerance in creeping bentgrass through analyzing various metabolites and enzymes involved in biosynthesis and catabolism of these metabolites. As you mentioned, genetic approaches could be used for analyzing changes in metabolic pathways. However, changes in metabolites and enzyme activities are also widely used to evaluate alterations in metabolic pathways.
Reviewer’s comment: I agree with the author’s reply. However, I think they still need to be careful in the way they “formulate” their conclusions, as changes in metabolites and enzymes activities is not sufficient enough to draw conclusions as they did. They can only speculate on the mechanisms involved, and emit hypotheses.
- The authors do not provide information about the internal standards they used, for the determination of the endogenous PAs content in the Material & methods.
Reply: Thank you very much for your careful review. Internal standards we used for determination of endogenous PAs have been added in line 454-456.
- Section 5.6 of the Material and Methods do not present the determination of the N and GABA metabolism as entitled.
Reply: Thanks. It was a typing error. The title of 5.6 has been revised to “5.6. Statistical Analyses” (line 494).
Author Response
Reviewer’s comment:
- The experimental design is better justified now. Minor English editing is required. I would suggest the authors to specify the number of plants included in their biological replicates. Is n=4 plants considered as four biological replicates? Or did they use a pool of n plants per biological replicate, and they reproduce the experiment four times, independently? This must be specified to assess the statistics properly.
Reply: Thank you very much for your professional and valuable review again. The whole manuscript has been checked again and the English has been improved carefully. The “n=4” indicated four biological replicates, but not 4 plants. Each biological replicate included many plants, because all parameters were detected based on leaf weight. For example, the determination of free proline content was estimated by using 0.1 g of leaf samples, and the 0.1 g of leaf sample were collected from more than 4 plants. As you mentioned, we used a pool of n plants per biological replicate, and we reproduce the experiment four times, independently. These information in details has been added in the section of Materials and Methods (line 430-431) in revised manuscript according to suggestions. The sentence “The “n=4” indicates four independent biological replicates.” has been also added in each figure legend in revised manuscript.
- I agree with the author’s reply. However, I think they still need to be careful in the way they “formulate” their conclusions, as changes in metabolites and enzymes activities is not sufficient enough to draw conclusions as they did. They can only speculate on the mechanisms involved, and emit hypotheses.
Reply: Thank you very much for your good suggestions, and we also absolutely agree with your suggestions. We only can speculate on the mechanisms involved in, and emit hypotheses. Some changes have been made in revised manuscript according to suggestions (line 366-368).